# MULTIPLE CORRELATION ENCODER-BASED NAIVE BAYES

## ABSTRACT

Naive Bayes (NB) continues to be one of the top 10 data mining algorithms. However, due to its assumption of attribute conditional independence, NB encounters significant challenges in addressing attribute-class correlations, attribute-attribute correlations, instance-class correlations, instance-instance correlations, and so on. In the last few decades, a large number of improved algorithms have been proposed, but none of them simultaneously addresses all these correlations. To bridge this gap, this paper proposes a novel algorithm called multiple correlation encoder-based naive Bayes (MCENB). In MCENB, we first design a multiple correlation encoder to generate new attributes, where multiple correlations are simultaneously captured and optimized. Specifically, the newly generated attributes are highly correlated with the class, yet uncorrelated with each other. Instances consisting of new attribute values are highly correlated with those in the same class. Subsequently, we augment original attributes by concatenating them with new attributes. Finally, we weight each augmented attribute to alleviate the attribute redundancy and then build NB on the weighted attributes. The experiments across numerous datasets show that MCENB significantly outperforms its benchmark competitors.

## 1 INTRODUCTION

Bayesian network classifiers (BNCs) (Pearl, 1989; Friedman et al., 1997; Tang et al., 2016; Zhang et al., 2020) have received much attention in the supervised classification due to their obvious advantages, such as the explicit interpretability and the powerful expression ability. In a supervised classification task containing $m$ attribute variables, an instance $\boldsymbol{x}$ can be represented as an attribute value vector $< a_1, a_2, ..., a_j, ..., a_m >$, where $a_j$ is the value of $\boldsymbol{x}$ on the $j$-th attribute $A_j$. Let $C$ represent the class variable and $c$ represent the value that $C$ takes, BNCs use Eq. (1) to classify $\boldsymbol{x}$:

$$\hat{c}(\boldsymbol{x}) = \underset{c \in C}{arg\,max}\, P(c)P(a_1, a_2, ..., a_j, ..., a_m|c), \tag{1}$$

where $\hat{c}(\boldsymbol{x})$ is the class label of $\boldsymbol{x}$ predicted by BNCs and $P(c)$ is the prior probability of $c$. Among numerous BNCs, naive Bayes (NB) (Settouti et al., 2016; Domingos & Pazzani, 1996) requires an assumption that all attributes are independent given the class, simplifying the probability estimation of $P(a_1, a_2, ..., a_j, ..., a_m|c)$. According to the assumption, NB uses Eq. (2) to classify $\boldsymbol{x}$:

$$\hat{c}(\boldsymbol{x}) = \underset{c \in C}{arg\,max}\, P(c)\prod_{j=1}^{m} P(a_j|c), \tag{2}$$

where $P(a_j|c)$ is the conditional probability of $a_j$ given $c$.

Although NB has continued to be one of the top 10 algorithms in data mining (Wu et al., 2008), it is obvious that the assumption of attribute conditional independence is rarely true since various correlations in the real world result in complex dependencies among attributes. To address these correlations, including attribute-class correlations, attribute-attribute correlations, instance-class correlations, instance-instance correlations, and so on, numerous improved algorithms of NB have been proposed. They can be broadly divided into four categories: structure-oriented, probability-oriented, attribute-oriented and instance-oriented algorithms.

Existing improved algorithms have demonstrated significant effectiveness, however, most of them focus on a single correlation, and none of them address all these correlations simultaneously. To

bridge this gap, this paper proposes a novel algorithm called multiple correlation encoder-based naive Bayes (MCENB). In MCENB, we first design a multiple correlation encoder to generate new attributes, where multiple correlations are simultaneously captured and optimized. Specifically, the newly generated attributes are highly correlated with the class, yet uncorrelated with each other. Instances consisting of new attribute values are highly correlated with those in the same class. Subsequently, we augment original attributes by concatenating them with new attributes. Finally, we weight each augmented attribute to alleviate the attribute redundancy and then build NB on the weighted attributes. In summary, the main contributions of this paper can be highlighted as:

- We argue that multiple correlations should be simultaneously optimized to improve NB. Attributes should be highly correlated with the class, yet uncorrelated with each other, and instances in the same class should be highly correlated with each other.
- We design a new multiple correlation encoder (MCE), which can capture multiple correlations and then optimize these correlations to generate new attributes with higher identification abilities compared to original attributes.
- We propose a novel algorithm called multiple correlations encoder-based naive Bayes (MCENB), which uses the original attributes and the new attributes and thus provides a more comprehensive attribute representation for improving NB.

The rest of this paper is organized as: Section 2 conducts a survey on improved algorithms of NB. Section 3 provides a description of our proposed MCENB. Section 4 conducts experiments on real-world and synthetic datasets. Section 5 concludes this paper and discusses the future work.

## 2 RELATED WORK

To address various correlations, numerous improved algorithms of NB have been proposed, which can be broadly divided into four categories: structure-oriented, probability-oriented, attribute-oriented and instance-oriented algorithms. Structure-oriented algorithms (Friedman et al., 1997; Webb et al., 2005; Jiang et al., 2009; Qiu et al., 2015) extend the network structure of NB to capture attribute-attribute correlations. They add directed edges among attribute vertices to represent attribute dependencies and calculate the conditional probabilities of attributes given both the class vertex and their parent vertices. Probability-oriented algorithms (Hindi, 2014; Diab & Hindi, 2017; Hindi et al., 2020; Zhang & Jiang, 2022) suggest that it is rough to represent attribute-class correlations by the conditional probabilities estimated by NB. Therefore, they claim to iteratively fine tune the conditional probabilities to obtain a more effective correlation representation. Attribute-oriented algorithms include three strategies: attribute generation, attribute selection and attribute weighting. Attribute generation (Ou et al., 2022; He et al., 2023) maps original attributes to another attribute space to reduce attribute-attribute correlations. Attribute selection (Chen et al., 2014; 2020) evaluates attribute-class correlations to remove attributes uncorrelated with the class. Attribute weighting (Zaidi et al., 2013; Jiang et al., 2019) evaluates attribute-class correlations and possibly attribute-attribute correlations to assign different weights for different attributes. Similar to attribute-oriented algorithms, instance-oriented algorithms also include three strategies: instance generation, instance selection and instance weighting. Specifically, instance generation (Jiang & Zhang, 2005; Jiang et al., 2008; Li et al., 2025) models instance-instance correlations to generate new instances. Instance selection (Langley & Sage, 2013; Wang et al., 2015) evaluates instance-class correlations to remove redundant and noisy instances in each class. Instance weighting (Jiang et al., 2012; 2014) evaluates instance-class correlations to assign different weights for different instances.

Based on the above discussion, we summarize the correlations addressed in different improved algorithms in Table 1. Existing improved algorithms focus on attribute-class correlations, attribute-attribute correlations, instance-class correlations and instance-instance correlations, but most address one of these correlations. To address the above four correlations simultaneously, this paper proposes a novel algorithm called multiple correlation encoder-based naive Bayes (MCENB).

## 3 MCENB

In this paper, we first design a new multiple correlation encoder (MCE), which can generate new attributes by capturing and optimizing multiple correlations. Based on the MCE, MCENB leverages

Table 1: Comparisons of correlations addressed in different improved algorithms of NB.

| Algorithm | attribute-class | attribute-attribute | instance-class | instance-instance |
|---|---|---|---|---|
| structure-oriented algorithms | - | ✓ | - | - |
| probability-oriented algorithms | ✓ | - | - | - |
| attribute-oriented algorithms | ✓ | ✓ | - | - |
| instance-oriented algorithms | - | - | ✓ | ✓ |
| our proposed MCENB | ✓ | ✓ | ✓ | ✓ |

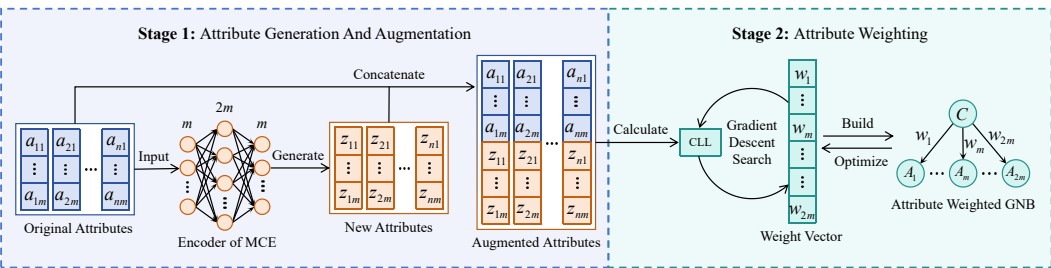

Figure 1: Framework of MCENB.

attribute generation, attribute augmentation and attribute weighting to improve NB. The framework of MCENB is illustrated in Figure 1. From Figure 1, we can see that MCENB is a two-stage algorithm. In the first stage, we generate new attributes by inputting original attributes to the encoder of MCE and then augment original attributes by concatenating them with new attributes. In the second stage, we optimize the weight vector by using the gradient descent search to maximize the conditional log-likelihood (CLL) and finally build attribute weighted Gaussian naive Bayes (GNB) on the augmented attributes.

## 3.1 ATTRIBUTE GENERATION AND AUGMENTATION

Given a training set with $n$ instances, $m$ attributes and $k$ classes, it can be represented as $\mathcal{D} = \{\mathcal{X}, \boldsymbol{c}\}$, where $\mathcal{X}$ is the set of $n$ original attribute value vectors, $\boldsymbol{c}$ is the class label vector. The $i$-th attribute value vector in $\mathcal{X}$ is $\boldsymbol{x}_i$, the $j$-th attribute value of $\boldsymbol{x}_i$ is $a_{ij}$ and the class label of $\boldsymbol{x}_i$ is $c_i$. For the given $\mathcal{D}$, we design MCE to generate a set of embedding vectors and treat them as new attribute value vectors. The structure of MCE is illustrated in Figure 2. From Figure 2, we can see that MCE consists of an encoder $q(\boldsymbol{z}|\boldsymbol{x})$ and a decoder $p(\boldsymbol{x}|\boldsymbol{z}, \boldsymbol{v})$, where $\boldsymbol{z}$ is the embedding vector generated by $q(\boldsymbol{z}|\boldsymbol{x})$ and $\boldsymbol{v}$ is the one-hot class vector. $q(\boldsymbol{z}|\boldsymbol{x})$ and $p(\boldsymbol{x}|\boldsymbol{z}, \boldsymbol{v})$ are both set to consist of three layers of neurons, with the number of neurons in each layer are $m$, $2m$, $m$ and $m+k$, $2m$, $m$, respectively. During the training process of MCE, original attribute value vectors are processed sequentially. Taking $\boldsymbol{x}_i$ as an example, it is first input to $q(\boldsymbol{z}|\boldsymbol{x})$ to generate an embedding vector $\boldsymbol{z}_i = <z_{i1}, z_{i2}, ..., z_{ij}, ..., z_{im}>$, where $z_{ij}$ is the $j$-th value of $\boldsymbol{z}_i$. Subsequently, to introduce the supervised information into the structure, $c_i$ is transformed into a one-hot class vector $\boldsymbol{v}_i$ and then $\boldsymbol{v}_i$ is concatenated with $\boldsymbol{z}_i$. Finally, the concatenated vector is input to $p(\boldsymbol{x}|\boldsymbol{z}, \boldsymbol{v})$ to generate a reconstructed attribute value vector $\hat{\boldsymbol{x}}_i$, where $\hat{a}_{ij}$ is the $j$-th reconstructed attribute value of $\hat{\boldsymbol{x}}_i$. Motivated by variational inference (Zhang et al., 2019), MCE is trained by maximizing the evidence lower bound (ELBO), which can be formulated as Eq. (3):

$$\mathcal{L}_{ELBO} = \mathbb{E}_{q(\boldsymbol{z}|\boldsymbol{x})}[\log p(\boldsymbol{x}|\boldsymbol{z}, \boldsymbol{v})] - KL(\mathcal{Z}||p(\mathcal{Z})), \qquad (3)$$

where $\mathbb{E}_{q(\boldsymbol{z}|\boldsymbol{x})}[\log p(\boldsymbol{x}|\boldsymbol{z}, \boldsymbol{v})]$ is the expectation of the log-likelihood about original attribute value vectors and reconstructed attribute value vectors, $KL(\mathcal{Z}||p(\mathcal{Z}))$ is the Kullback-Leibler divergence between the set of embedding vectors $\mathcal{Z}$ and the variational prior $p(\mathcal{Z})$. We treat the $i$-th embedding vector $\boldsymbol{z}_i$ in $\mathcal{Z}$ as the $i$-th new attribute value vector, i.e., the $i$-th new instance. Meanwhile, we define the $j$-th new attributes as $B_j$, and $p(\mathcal{Z})$ requires that each new attribute follows a standard Gaussian distribution given the class.

Based on the structure of MCE, we further improve the identification abilities of new attributes by capturing and optimizing multiple correlations from two perspectives, which are illustrated in Figure 3. From Figure 3, we can see that attribute-class correlations and attribute-attribute correlations

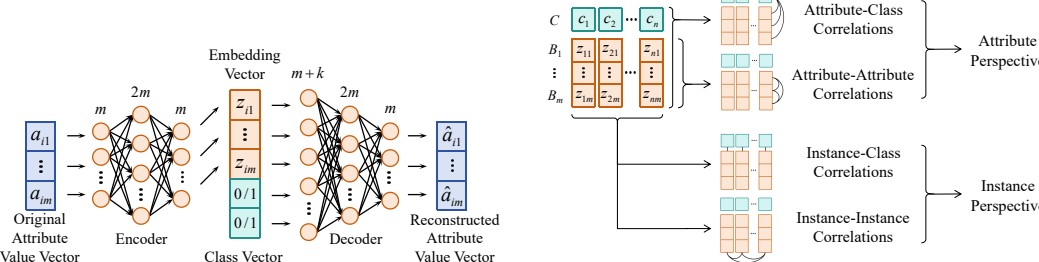

Figure 2: Structure of MCE.

Figure 3: Correlations about new attributes.

can be optimized from the attribute perspective, as well as instance-class correlations and instance-instance correlations from the instance perspective. With these two perspectives, we introduce two new correlation objective functions: the attribute correlation function $\mathcal{L}_A$ and the instance correlation function $\mathcal{L}_I$. In this paper, the correlation $\rho(\boldsymbol{\alpha}; \boldsymbol{\beta})$ between two vectors $\boldsymbol{\alpha}$ and $\boldsymbol{\beta}$ is captured by the Pearson coefficient (Cohen & Israel, 2009). The sign of the Pearson coefficient is neglected, as it is uncorrelated with our analysis. Therefore, $\rho(\boldsymbol{\alpha}; \boldsymbol{\beta})$ can be formulated as Eq. (4):

$$\rho(\boldsymbol{\alpha}; \boldsymbol{\beta}) = \frac{\left| \sum_{l'=1}^{l} (\alpha_{l'} - \mu_\alpha)(\beta_{l'} - \mu_\beta) \right|}{\sqrt{\sum_{l'=1}^{l} (\alpha_{l'} - \mu_\alpha)^2} \sqrt{\sum_{l'=1}^{l} (\beta_{l'} - \mu_\beta)^2}}, \tag{4}$$

where $l$ is the length of $\boldsymbol{\alpha}$ and $\boldsymbol{\beta}$, $\alpha_{l'}$ and $\beta_{l'}$ are the $l'$-th element values of $\boldsymbol{\alpha}$ and $\boldsymbol{\beta}$, respectively, $\mu_\alpha$ and $\mu_\beta$ are the means of element values in $\boldsymbol{\alpha}$ and $\boldsymbol{\beta}$, respectively.

**Attribute perspective.** Let $\boldsymbol{b}_j = <z_{1j}, z_{2j}, ..., z_{ij}, ..., z_{nj}>$ represent the vector consisting of the attribute values taken by $B_j$ in all new instances, the attribute-class correlation between $\boldsymbol{b}_j$ and $\boldsymbol{c}$ is represented as $\rho(\boldsymbol{b}_j; \boldsymbol{c})$. Similarly, the attribute-attribute correlation between $\boldsymbol{b}_j$ and $\boldsymbol{b}_u$ is represented as $\rho(\boldsymbol{b}_j; \boldsymbol{b}_u)$. To align with the assumption of attribute conditional independence, new attributes should be highly correlated with the class yet uncorrelated with each other. Therefore, we calculate the average attribute-class correlation $\bar{\rho}_{\boldsymbol{bc}}$ and the average attribute-attribute correlation $\bar{\rho}_{\boldsymbol{bb}}$ of new attributes, and then enhance the difference between $\bar{\rho}_{\boldsymbol{bc}}$ and $\bar{\rho}_{\boldsymbol{bb}}$ by maximizing the attribute correlation function $\mathcal{L}_A$, which is formulated as Eq. (5):

$$\mathcal{L}_A = \bar{\rho}_{\boldsymbol{bc}} - \bar{\rho}_{\boldsymbol{bb}} = \frac{1}{m} \sum_{j=1}^{m} \rho(\boldsymbol{b}_j; \boldsymbol{c}) - \frac{1}{m(m-1)} \sum_{j=1}^{m} \sum_{u=1 \wedge u \neq j}^{m} \rho(\boldsymbol{b}_j; \boldsymbol{b}_u). \tag{5}$$

**Instance perspective.** The instance-instance correlation between $\boldsymbol{z}_i$ and $\boldsymbol{z}_v$ is $\rho(\boldsymbol{z}_i; \boldsymbol{z}_v)$. Since the class label of each instance is a scalar rather than a vector, instance-class correlations can not be explicitly measured by the Pearson coefficient. In this context, we indirectly optimize instance-class correlations through instance-instance correlations. Specifically, new instances in the same class should be highly correlated with each other to tightly aggregate together, thereby enhancing the correlation between each instance and its class. Following the concept of contrastive learning (Khosla et al., 2020), we enhance the ratio between the average instance-instance correlation of instances in each class and the average instance-instance correlation of all instances. This goal is achieved by maximizing the instance correlation function $\mathcal{L}_I$, which can be formulated as Eq. (6):

$$\mathcal{L}_I = \frac{1}{k} \sum_{c=1}^{k} r_c = \frac{1}{k} \sum_{c=1}^{k} \frac{\frac{1}{n_c(n_c-1)} \sum_{i=1}^{n_c} \sum_{v=1 \wedge v \neq i}^{n_c} \rho(\boldsymbol{z}_i; \boldsymbol{z}_v)}{\frac{1}{n(n-1)} \sum_{i=1}^{n} \sum_{v=1 \wedge v \neq i}^{n} \rho(\boldsymbol{z}_i; \boldsymbol{z}_v)}, \tag{6}$$

where $r_c$ is the instance-instance correlation ratio corresponding to the $c$-th class, and $n_c$ is the number of instances in the $c$-th class.

Considering the ELBO and two correlation objective functions, we finally maximize $\mathcal{L}$ to train MCE, which can be formulated as Eq. (7):

$$\mathcal{L} = \mathcal{L}_{ELBO} + \mathcal{L}_A + \mathcal{L}_I, \tag{7}$$

based on $\mathcal{L}$, we update the parameters $\mathcal{W}$ of MCE over $P$ iterations by Eq. (8):

$$\mathcal{W}_{p+1} = \mathcal{W}_p - \eta \frac{\partial \mathcal{L}}{\partial \mathcal{W}}, \tag{8}$$

where $\mathcal{W}_p$ represents the parameters in the $p$-th iteration and $\eta$ represents the learning rate. When $p = 1$, $\mathcal{W}_p$ represents randomly initialized parameters.

Finally, we use the MCE trained after $P$ iterations to generate new attributes. Since MCE optimizes multiple correlations from the attribute and the instance perspectives simultaneously, new attributes exhibit higher identification abilities compared to original attributes. To enhance the identification abilities of original attributes and provide a more comprehensive attribute representation, we augment original attributes by concatenating them with new attributes. For each instance, we define $z_{ij}$ as the $(m + j)$-th attribute value of $\boldsymbol{x}_i$. The length of $\boldsymbol{z}_i$ is the same as that of $\boldsymbol{x}_i$, it is evident that each instance can obtain $m$ new attributes, with the final length expanding to $2m$.

## 3.2 ATTRIBUTE WEIGHTING

Since new attributes are generated from original attributes, attribute augmentation may potentially cause the attribute redundancy. In this subsection, to alleviate the impact of attribute redundancy, we assign different weights for different augmented attributes. Besides, to align with the numerical attributes generated by MCE, we employ GNB to predict the class label, which is a variant of NB specifically adapted to numerical attributes. Based on the weighted attributes, we build attribute weighted GNB and use Eq. (9) to predict the class label:

$$\hat{c}(\boldsymbol{x}) = \arg\max_{c \in C} P(c) \prod_{j=1}^{2m} P(a_j|c)^{w_j}, \tag{9}$$

where $w_j$ is the weight of $A_j$. $P(c)$ and $P(a_j|c)$ are estimated by Eq. (10) and Eq. (11), respectively:

$$P(c) = \frac{\sum_{i=1}^{n} I(c_i, c) + 1}{n + k}, \tag{10}$$

$$P(a_j|c) = \frac{1}{\sqrt{2\pi}\sigma_{cj}} \exp\left(-\frac{(a_j - \mu_{cj})^2}{2\sigma_{cj}^2}\right), \tag{11}$$

where $I(\cdot)$ is a binary function, which takes the value 1 if its two parameters are identical and 0 otherwise. $\mu_{cj}$ and $\sigma_{cj}$ are the mean and standard deviation of $A_j$ given $c$, respectively.

After building the attribute weighted GNB, we assign different weights for different augmented attributes. Initially, each weight in the weight vector $\boldsymbol{w}$ is assigned a random value between 0 and 1. Subsequently, these weights are optimized by the gradient descent search. The objective function of optimization is defined to maximize the CLL of the attribute weighted GNB, which can be formulated as Eq. (12):

$$\text{CLL}(\boldsymbol{w}) = \log \hat{P}(C|\boldsymbol{\mathcal{D}}, \boldsymbol{w}) = \sum_{i=1}^{n} \log \hat{P}(c_i|\boldsymbol{x}_i, \boldsymbol{w}), \tag{12}$$

where $\hat{P}(c_i|\boldsymbol{x}_i, \boldsymbol{w})$ is the posterior probability of $c_i$ estimated by the attribute weighted GNB given $\boldsymbol{x}_i$ and $\boldsymbol{w}$, which is formulated as Eq. (13):

$$\hat{P}(c_i|\boldsymbol{x}_i, \boldsymbol{w}) = \frac{\varphi(c_i|\boldsymbol{x}_i, \boldsymbol{w})}{\sum\limits_{c=1}^{k} \varphi(c|\boldsymbol{x}_i, \boldsymbol{w})}, \tag{13}$$

where $\varphi(c|\boldsymbol{x}_i, \boldsymbol{w})$ is the product of $P(c)$ and each $P(a_j|c)^{w_j}$ of $\boldsymbol{x}_i$, which is formulated as Eq. (14):

$$\varphi(c|\boldsymbol{x}_i, \boldsymbol{w}) = P(c) \prod_{j=1}^{2m} P(a_j|c)^{w_j}. \tag{14}$$

The gradient of $\varphi(c|\boldsymbol{x}_i, \boldsymbol{w})$ with respect to $w_j$ can be formulated as Eq. (15):

$$
\begin{aligned}
\frac{\partial}{\partial w_j} \varphi(c|\boldsymbol{x}_i, \boldsymbol{w}) &= \left( P(c) \prod_{j'=1 \wedge j' \neq j}^{2m} P(a_{j'}|c)^{w_{j'}} \right) \frac{\partial}{\partial w_j} P(a_j|c)^{w_j} \\
&= \left( P(c) \prod_{j'=1 \wedge j' \neq j}^{2m} P(a_{j'}|c)^{w_{j'}} \right) P(a_j|c)^{w_j} \log P(a_j|c) \\
&= \varphi(c|\boldsymbol{x}_i, \boldsymbol{w}) \log P(a_j|c).
\end{aligned}
\tag{15}
$$

Then, the gradient of $\mathrm{CLL}(\boldsymbol{w})$ with respect to $w_j$ can be formulated as Eq. (16):

$$
\begin{aligned}
\frac{\partial}{\partial w_j} \mathrm{CLL}(\boldsymbol{w}) &= \frac{\partial}{\partial w_j} \sum_{i=1}^{n} \left( \log \varphi(c_i|\boldsymbol{x}_i, \boldsymbol{w}) - \log \left( \sum_{c=1}^{k} \varphi(c|\boldsymbol{x}_i, \boldsymbol{w}) \right) \right) \\
&= \sum_{i=1}^{n} \left( \log P(a_j|c_i) - \frac{\sum_{c=1}^{k} \varphi(c|\boldsymbol{x}_i, \boldsymbol{w}) \log P(a_j|c)}{\sum_{c=1}^{k} \varphi(c|\boldsymbol{x}_i, \boldsymbol{w})} \right) \\
&= \sum_{i=1}^{n} \left( \log P(a_j|c) - \sum_{c=1}^{k} \hat{P}(c|\boldsymbol{x}_i, \boldsymbol{w}) \log P(a_j|c) \right).
\end{aligned}
\tag{16}
$$

In summary, the entire learning algorithm for our MCENB can be partitioned into training (MCENB-training) and classification (MCENB-classification) algorithms. These two algorithms and their time complexity analyses are provided in **Appendix** A and **Appendix** B, respectively.

## 4 EXPERIMENTS AND RESULTS

The purpose of this section is to validate the effectiveness and rationality of our proposed MCENB. We first compare the classification accuracy of MCENB with its seven competitors and conduct two groups of ablation studies on 24 real-world datasets. Then, we observe the identification abilities of new attributes and the correlation optimization process on a synthetic dataset.

### 4.1 EXPERIMENTS ON 24 REAL-WORLD DATASETS

From the real-world datasets published by the KEEL dataset repository,[1] we use the whole 24 datasets only containing numerical attributes, which represent a wide range of domains and data characteristics. The detailed descriptions of these datasets are provided in **Appendix** C. In our experiments, attributes are normalized by the min-max method (Patro & Sahu, 2015).[2] We compare the classification accuracy of MCENB with its seven competitors on these datasets by running 10 separate hold-out validations, in which each dataset is split into a training set and a test set by stratified sampling. The proportions of the training set and the test set are 80% and 20%, respectively.

**Classification performance.** To verify the classification performance of our proposed MCENB, we compare it with GNB and its six state-of-the-art competitors, including attribute grouping-based naive Bayesian classifier (AG-NBC) (He et al., 2023), auto-encoding naive Bayesian classifier (AE-NBC) (Ou et al., 2022), weighting attributes to alleviate naive Bayes' independence assumption (WANBIA) (Zaidi et al., 2013), correlation-based feature weighting filter for naive Bayes (CFWNB) (Jiang et al., 2019), instance correlation graph-based niave Bayes (ICGNB) (Li et al., 2025) and discriminatively weighted naive Bayes (DWNB) (Jiang et al., 2012). Among all competitors, GNB is

---

[1]https://sci2s.ugr.es/keel/category.php?cat=clas

[2]Some other normalization methods, such as the z-score method, can not limit the range of attribute values, which will lead to the computational overflow of $\mathcal{L}_{ELBO}$.

Table 2: Classification accuracy (%) comparisons for MCENB versus its competitors.

| Dataset | MCENB | AG-NBC | AE-NBC | WANBIA | CFWNB | ICGNB | DWNB | GNB |
|---|---|---|---|---|---|---|---|---|
| appendicitis | 88.64±7.40 | 84.55±5.45 | 84.55±8.67 | 88.18±7.66 | **90.00±6.03** | 88.18±6.17 | **90.00±6.03** | 86.36±6.10 |
| balance | 89.76±1.75 | 91.04±1.78 | 87.44±5.20 | 90.24±2.84 | 88.32±3.87 | 86.32±2.74 | **91.36±2.23** | 90.24±2.84 |
| banana | 64.81±1.37 | **84.35±4.53** | 60.48±1.17 | 62.00±0.91 | 59.74±1.48 | 62.90±1.11 | 61.46±0.96 | 62.00±0.91 |
| cleveland | **58.33±4.41** | 53.83±4.09 | 56.50±7.94 | 58.00±4.27 | 54.83±4.86 | 54.83±4.37 | 55.50±7.19 | 52.17±8.95 |
| ecoli | **79.71±5.09** | 68.38±5.47 | 74.12±4.42 | 79.12±6.06 | 60.59±11.96 | 77.65±5.76 | 62.65±6.52 | 60.74±6.61 |
| glass | **61.16±7.86** | 59.07±6.91 | 54.88±9.93 | 59.07±7.15 | 51.63±11.34 | 58.37±8.02 | 47.67±9.14 | 47.21±9.70 |
| iris | 96.00±3.27 | 90.67±4.16 | 95.00±4.53 | 96.00±3.27 | 95.33±3.71 | **96.33±3.14** | **96.33±3.14** | 95.33±3.71 |
| led7digit-01 | **73.80±1.89** | 69.20±3.76 | 71.30±2.83 | 70.40±5.90 | 64.20±9.11 | 72.30±3.61 | 61.10±15.00 | 63.40±12.11 |
| magic | 77.17±0.67 | 75.74±2.09 | 76.52±0.48 | 77.08±0.56 | 74.89±0.44 | **80.39±0.50** | 79.46±0.52 | 72.66±0.64 |
| movement_libras | 68.89±3.84 | 70.83±4.35 | **76.53±5.47** | 63.06±4.90 | 62.22±4.76 | 58.21±5.23 | 62.50±5.27 | 61.94±5.63 |
| phoneme | 77.04±1.40 | 77.03±1.51 | 76.57±2.74 | 75.91±1.40 | 76.85±1.58 | 76.73±1.59 | **77.74±1.45** | 75.97±1.65 |
| pima | 75.65±2.83 | 73.38±3.17 | 72.21±3.14 | 75.52±2.69 | 75.00±2.97 | **75.97±1.86** | 75.91±2.94 | 74.61±3.45 |
| ring | 97.89±0.24 | 93.03±1.52 | 97.84±0.37 | 97.90±0.20 | 97.95±0.30 | **97.95±0.25** | 97.89±0.28 | 97.92±0.28 |
| segment | 89.46±1.14 | 87.97±1.32 | 82.49±1.04 | 88.79±1.27 | 14.39±1.54 | **89.76±1.13** | 85.87±1.12 | 79.42±1.48 |
| sonar | **83.33±5.32** | 74.76±8.86 | 66.67±6.39 | 78.33±5.05 | 68.10±5.02 | 72.14±4.40 | 78.33±4.70 | 66.67±5.11 |
| spambase | **92.97±0.98** | 87.59±1.40 | 86.78±0.96 | 90.26±1.14 | 83.73±1.25 | 91.04±1.13 | 83.42±2.28 | 82.08±1.25 |
| texture | 91.84±0.88 | **95.35±0.75** | 94.46±0.64 | 84.47±1.00 | 78.35±1.38 | 91.62±0.83 | 78.38±1.21 | 77.45±1.39 |
| titanic | 77.51±1.23 | 74.44±1.97 | 76.76±0.99 | 77.64±1.21 | 76.98±0.89 | 77.64±1.21 | **78.07±1.32** | 76.98±0.89 |
| twonorm | 97.72±0.26 | 96.20±0.93 | 96.92±0.78 | 97.72±0.27 | 97.71±0.29 | 97.64±0.28 | **97.76±0.32** | 97.70±0.28 |
| wdbc | 96.93±1.32 | 93.86±2.69 | 87.72±2.15 | 96.40±1.54 | 93.95±1.90 | **97.02±1.53** | 95.79±1.56 | 92.98±2.29 |
| wine | 97.78±1.11 | **98.06±2.50** | 93.89±2.72 | 97.50±1.50 | 96.94±1.94 | 97.50±1.50 | 96.67±2.42 | 97.50±1.50 |
| winequality-red | **59.56±2.11** | 58.84±3.23 | 52.06±3.08 | 58.44±1.78 | 58.53±1.48 | 58.97±2.00 | 55.50±2.47 | 54.72±2.56 |
| winequality-white | **53.19±1.06** | 50.97±2.06 | 45.62±2.56 | 52.21±1.51 | 49.27±1.16 | 52.41±0.92 | 47.69±1.58 | 44.38±1.61 |
| yeast | **56.90±3.10** | 51.01±4.60 | 42.93±7.67 | 54.28±3.20 | 18.86±4.12 | 55.59±3.84 | 15.69±2.49 | 14.92±3.24 |
| (W / T / L) | | 19/0/5 | 22/0/2 | 19/2/3 | 22/0/2 | 17/0/7 | 15/0/9 | 22/0/2 |
| **Average** | 79.42 | 77.51 | 75.43 | 77.86 | 73.10 | 77.81 | 73.86 | 71.89 |

Table 3: Wilcoxon tests for MCENB versus its competitors.

| Algorithm | MCENB | AG-NBC | AE-NBC | WANBIA | CFWNB | ICGNB | DWNB | GNB |
|---|---|---|---|---|---|---|---|---|
| MCENB | - | ○ | ○ | ○ | ○ | ○ | ○ | ○ |
| AG-NBC | ● | - | | ● | ○ | | | ○ |
| AE-NBC | ● | | - | ● | | ● | | ○ |
| WANBIA | ● | | ○ | - | ○ | | ○ | ○ |
| CFWNB | ● | ● | | ● | - | | ● | ○ |
| ICGNB | ● | | ○ | | ○ | - | | ○ |
| DWNB | ● | | | ● | | | - | ○ |
| GNB | ● | ● | ● | ● | ● | ● | ● | - |

the baseline. AG-NBC and AE-NBC address attribute-attribute correlations through attribute generation. WANBIA addresses attribute-class correlations through attribute weighting. CFWNB addresses both attribute-class correlations and attribute-attribute correlations through attribute weighting. ICGNB addresses instance-instance correlations through instance generation. DWNB addresses instance-class correlations through instance weighting. For algorithms focusing on nominal attributes (WANBIA, CFWNB and DWNB), we replace NB in them with GNB. In MCENB, the number of iterations $P$ is set to 200 and the learning rate $\eta$ is set to 0.01. The parameters of competitors are consistent with those in the original papers.

Table 2 shows the detailed classification accuracy (%) of each algorithm on each dataset. In Table 2, the highest classification accuracy on each dataset is bolded. The Win / Tie / Lose (**W / T / L**) values and the averages (arithmetic mean) are summarized at the bottom of the table. The Win/ Tie/ Lose values imply that MCENB wins on **W** datasets, ties on **T** datasets, and loses on **L** datasets compared to its competitor. Based on the classification accuracy results in Table 2, we employ the Wilcoxon signed-ranks test (Demsar, 2006) to conduct a comprehensive comparison of each pair of algorithms, and the comparison results are summarized in Table 3. In Table 3, symbol ● indicates that the algorithm in the column outperforms the one in the corresponding row, and symbol ○ signifies the opposite. The lower-diagonal significance level is $\alpha = 0.05$, while the upper-diagonal level is $\alpha = 0.1$. Observing these results, the conclusion is evident that MCENB significantly outperforms all the other competitors. Specific conclusions are summarized as:

(1) Compared to AG-NBC, AE-NBC, WANBIA and CFWNB, MCENB is better on 19, 22, 19, 22 datasets and only worse on 5, 2, 3, 2 datasets, respectively. MCENB wins the competitors addressing the correlations from the attribute perspective. This verifies the effectiveness of optimizing the correlations from the instance perspective.

(2) Compared to ICGNB and DWNB, MCENB is better on 17, 15 datasets and worse on 7, 9 datasets. MCENB wins the competitors addressing the correlations from the instance perspective. This verifies the effectiveness of optimizing the correlations from the attribute perspective.

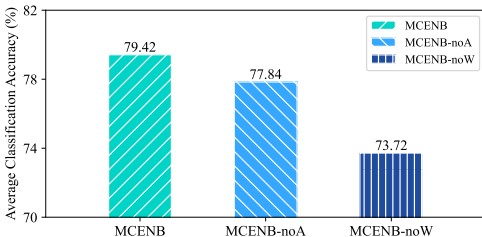 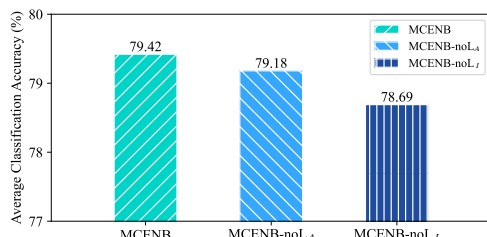

Figure 4: Average classification accuracy (%) of MCENB, MCENB-noA and MCENB-noW.

Figure 5: Average classification accuracy (%) of MCENB, MCENB-noL$_A$ and MCENB-noL$_I$.

(3) Compared to GNB, MCENB is better on 22 datasets and worse on 2 datasets. This indicates that MCENB is effective in enhancing the performance of GNB, and our proposed MCE is powerful.

(4) The average classification accuracy of MCENB (79.42%) is higher than those of AG-NBC (77.51%), AE-NBC (75.43%), WANBIA (77.86%), CFWNB (73.10%), ICGNB (77.81%), DWNB (73.86%) and GNB (71.89%), respectively. This suggests that MCENB performs better than existing state-of-the-art competitors.

(5) According to the Wilcoxon signed-ranks test results presented in Table 3, MCENB significantly outperforms all existing state-of-the-art competitors whether $\alpha = 0.05$ or $\alpha = 0.1$, which strongly validates the classification performance of MCENB.

**Ablation studies.** To validate the necessity of each stage in MCENB and each correlation objective function in MCE, we perform two groups of ablation studies by comparing MCENB with its ablation variants. The first group of variants are MCENB-noA and MCENB-noW. MCENB-noA removes the stage of attribute generation and augmentation while retaining attribute weighting. MCENB-noW removes the stage of attribute weighting while retaining attribute generation and augmentation. The second group of variants are MCENB-noL$_A$ and MCENB-noL$_I$. MCENB-noL$_A$ removes the correlation function $\mathcal{L}_A$ while retaining $\mathcal{L}_I$, and MCENB-noL$_I$ removes the correlation function $\mathcal{L}_I$ while retaining $\mathcal{L}_A$. Figures 4 and 5 show the average classification accuracy (%) of MCENB and these variants. Observing the results of MCENB, MCENB-noA and MCENB-noW in Figure 4, we summarize the conclusions as: (1) The accuracy of MCENB is higher than that of MCENB-noA, which confirms that enhancing the identification abilities of the original attributes through attribute generation and augmentation is necessary. (2) The accuracy of MCENB is higher than that of MCENB-noW, which confirms that alleviating the attribute redundancy through attribute weighting is necessary. Observing the results of MCENB, MCENB-noL$_A$ and MCENB-noL$_I$ in Figure 5, we summarize the conclusions as: (1) The accuracy of MCENB is higher than that of MCENB-noL$_A$, which confirms that optimizing the correlations from the attribute perspective is necessary. (2) The accuracy of MCENB is higher than that of MCENB-noL$_I$, which confirms that optimizing the correlations from the instance perspective is necessary.

### 4.2 EXPERIMENTS ON A SYNTHETIC DATASET

To observe the identification abilities of new attributes and the correlation optimization process, we design another group of experiments on a synthetic dataset, which contains 2 classes, 100 instances and 50 attributes. The synthetic process initially creates Gaussian clusters around the vertices of a 40-dimensional hypercube and assigns an equal number of clusters to each class. By sampling from these clusters, instances containing 40 informative attributes can be obtained. Then, 10 redundant attributes are generated by random linear combinations of these informative attributes.

**Identification abilities.** To observe the identification abilities of new attributes generated by MCE, we compare the class distributions of original attributes and new attributes. We employ the t-SNE algorithm (van der Maaten & Hinton, 2008) to map the dataset to a two-dimensional space, providing clear observations of the class distributions. Figure 6 shows the class distributions of original attributes and new attributes, where scatters with a circle are test instances and the rest are training instances. In Figure 6a, we can see that the class distribution of original attributes is confusing, and

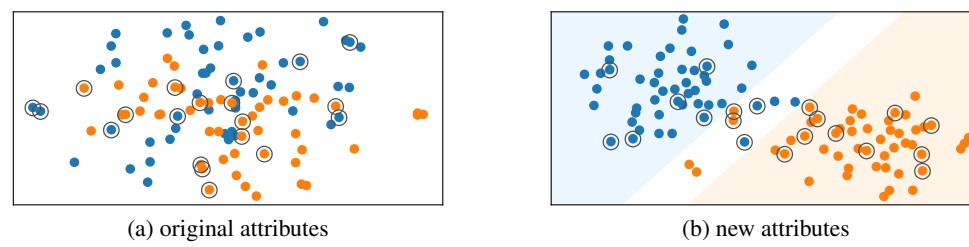

(a) original attributes      (b) new attributes

Figure 6: Class distributions of the synthetic dataset.

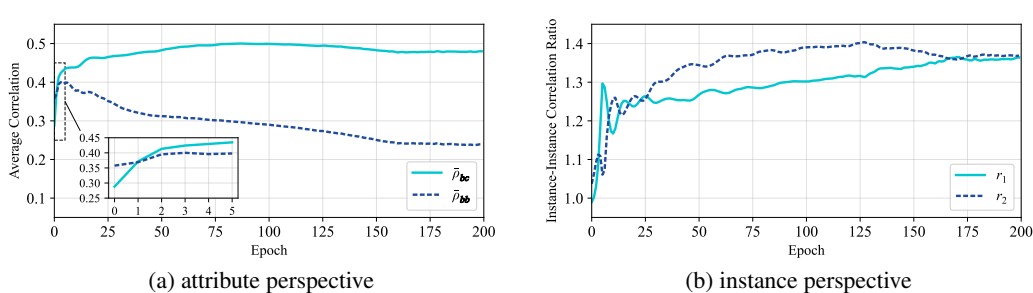

(a) attribute perspective      (b) instance perspective

Figure 7: Correlation changes during the MCE training process.

there is no class boundary between the two classes. In Figure 6b, we can see that the class distribution of new attributes is significantly more distinguishable compared to that of original attributes. Besides, most test instances are aggregated together with the training instances in the same class, while only a few test instances are in the class boundary. This facilitates predicting the class of each instance accurately. In summary, the class distributions in Figure 6 indicate that new attributes exhibit higher identification abilities compared to original attributes.

**Correlation optimization.** To observe how correlations are optimized, we show the correlation changes during the MCE training process in Figure 7. From the attribute perspective, Figure 7a shows the changes of $\bar{\rho}_{bc}$ and $\bar{\rho}_{bb}$. At the beginning, $\bar{\rho}_{bb}$ is higher than $\bar{\rho}_{bc}$, but $\bar{\rho}_{bc}$ exceeds it after only 2 iterations. As $\bar{\rho}_{bc}$ increases while $\bar{\rho}_{bb}$ decreases, the difference between them gradually grows, eventually stabilizing at approximately 0.2, which represents a significant rise compared to the initial state. This suggests that new attributes exhibit higher attribute-class correlations and lower attribute-attribute correlations compared to original attributes. From the instance perspective, Figure 7b shows the changes of $r_1$ and $r_2$, which are the instance-instance correlation ratios corresponding to two classes, respectively. At the beginning, they are approximately 1.0 and 1.05, respectively. After the training process, they both increase to approximately 1.35. This suggests that new attributes exhibit higher instance-instance correlations in the same class compared to original attributes. The correlation changes in Figure 7 indicate that multiple correlations about new attributes are optimized from both the attribute and the instance perspectives.

## 5 CONCLUSION AND FEATURE WORK

To obtain highly identifiable attributes, we design a new multiple correlation encoder (MCE) to generate new attributes by capturing and optimizing multiple correlations. Based on it, we propose a novel algorithm called multiple correlation encoder-based naive Bayes (MCENB). Comprehensive experiments demonstrate its effectiveness and rationality on both real-world and synthetic datasets. However, MCENB only considers the absolute value not the sign of the Pearson coefficient, failing to consider the negative correlation. Exploring how to leverage the negative correlation is the main direction for future work.

## REPRODUCIBILITY STATEMENT

We submit the code and datasets as supplementary materials, and the details of dataset preprocessing and algorithm implementation are provided in the main text. Once our paper is accepted, we will make the code and datasets publicly available on GitHub.

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

# Appendix A  Training algorithm of MCENB

---

**Algorithm 1** MCENB-training($\mathcal{D}$, $P$, $\eta$)

---

**Input**: $\mathcal{D} = \{\mathcal{X}, \boldsymbol{c}\}$ - the training set, $P$ - the number of iterations, $\eta$ - the learning rate.
**Output**: $\boldsymbol{w}$ - the weight vector, $q(\boldsymbol{z}|\boldsymbol{x})$ - the encoder of MCE.

 1: Initialize the parameters of MCE as $\mathcal{W}_1$;
 2: **for** $p = 1$ to $P$ **do**
 3:    **for** $i = 1$ to $n$ **do**
 4:       Generate the $i$-th embedding vector $\boldsymbol{z}_i$ by $q(\boldsymbol{z}|\boldsymbol{x})$ with $\mathcal{W}_p$;
 5:       Transform the class label of $c_i$ into the one-hot class vector $\boldsymbol{v}_i$ and concatenate $\boldsymbol{z}_i$ with it;
 6:       Generate the reconstructed attribute value vector $\hat{\boldsymbol{x}}_i$ by $p(\boldsymbol{x}|\boldsymbol{z}, \boldsymbol{v})$ with $\mathcal{W}_p$;
 7:    **end for**
 8:    Calculate the value of the objective function $\mathcal{L}$ by Eqs. (3)-(7);
 9:    Update the parameters of MCE with $\eta$ by Eq. (8);
10: **end for**
11: **for** $i = 1$ to $n$ **do**
12:    Generate the embedding vector $\boldsymbol{z}_i$ by $q(\boldsymbol{z}|\boldsymbol{x})$ with $\mathcal{W}_{P+1}$;
13:    Concatenate $\boldsymbol{x}_i$ with $\boldsymbol{z}_i$;
14: **end for**
15: **for** $c = 1$ to $k$ **do**
16:    Estimate the prior probability $P(c)$ by Eq. (10);
17:    **for** $j = 1$ to $2m$ **do**
18:       Estimate the conditional probability $P(a_j|c)$ by Eq. (11);
19:    **end for**
20: **end for**
21: Assign random values between 0 and 1 to weights in the weight vector $\boldsymbol{w}$;
22: Optimize the initialized weight vector $\boldsymbol{w}$ by Eqs. (12)-(16);
23: **return** $\boldsymbol{w}$, $q(\boldsymbol{z}|\boldsymbol{x})$.

---

In **Algorithm 1**, line 1 initializes the parameters in MCE with a time complexity of $O(m + k)$. Lines 2-10 train an MCE with a time complexity of $O(P(nm(m + k)) + n^2mk)$. Lines 11-14 generate new attributes and augment original attributes with a time complexity of $O(nm^2)$. Lines 15-20 train a GNB with a time complexity of $O(knm)$. Lines 21-22 weight augmented attributes with a time complexity of $O(\beta(m))$, where $\beta(m)$ has a linear relationship with $m$. Due to $n$ usually being greater than $m$ and $m$ usually being greater than $k$, considering only the highest-order terms, the overall time complexity of **Algorithm 1** is $O(Pn^2mk)$.

# Appendix B  Classification algorithm of MCENB

---

**Algorithm 2** MCENB-classification($\boldsymbol{x}$, $\boldsymbol{w}$, $q(\boldsymbol{z}|\boldsymbol{x})$)

---

**Input**: $\boldsymbol{x}$ - a test instance, $\boldsymbol{w}$ - the weight vector, $q(\boldsymbol{z}|\boldsymbol{x})$ - the encoder of MCE.
**Output**: $\hat{c}(\boldsymbol{x})$ - the predicted class label of $\boldsymbol{x}$.

 1: Generate the embedding vector $\boldsymbol{z}$ by $q(\boldsymbol{z}|\boldsymbol{x})$;
 2: Concatenate $\boldsymbol{x}$ with $\boldsymbol{z}$;
 3: **for** $c = 1$ to $k$ **do**
 4:    Estimate the prior probability $P(c)$ by Eq. (10);
 5:    **for** $j = 1$ to $2m$ **do**
 6:       Estimate the conditional probability $P(a_j|c)$ by Eq. (11);
 7:    **end for**
 8: **end for**
 9: Predict the class label $\hat{c}(\boldsymbol{x})$ of $\boldsymbol{x}$ by Eq. (9);
10: **return** $\hat{c}(\boldsymbol{x})$.

---

In **Algorithm 2**, lines 1-2 generate new attributes and augment original attributes with a time complexity of $O(m^2)$. Lines 3-8 estimate $P(c)$ and each $P(a_j|c)$ with a time complexity of $O(km)$. Line 9 predicts $\hat{c}(x)$ with a time complexity of $O(m)$. Considering only the highest-order terms, the overall time complexity of **Algorithm 2** is $O(m^2)$.

# Appendix C   Descriptions of 24 real-world datasets used in experiments

| Dataset | #Attributes | #Classes | #Instances |
|---|---|---|---|
| appendicitis | 7 | 2 | 106 |
| balance | 4 | 3 | 625 |
| banana | 2 | 2 | 5300 |
| cleveland | 13 | 5 | 297 |
| ecoli | 7 | 8 | 336 |
| glass | 9 | 7 | 214 |
| iris | 4 | 3 | 150 |
| led7digit-01 | 7 | 10 | 500 |
| magic | 10 | 2 | 19020 |
| movement_libras | 90 | 15 | 360 |
| phoneme | 5 | 2 | 5404 |
| pima | 8 | 2 | 768 |
| ring | 20 | 2 | 7400 |
| segement | 19 | 7 | 2310 |
| sonar | 60 | 2 | 208 |
| spambase | 57 | 2 | 4597 |
| texture | 40 | 11 | 5500 |
| titanic | 3 | 2 | 2201 |
| twonorm | 20 | 2 | 7400 |
| wdbc | 30 | 2 | 569 |
| wine | 13 | 3 | 178 |
| winequality-red | 11 | 11 | 1599 |
| winequality-write | 11 | 11 | 4898 |
| yeast | 8 | 10 | 1484 |

This table provides detailed descriptions of datasets used in our experiments, where #Attributes is the number of attributes, #Classes is the number of classes and #Instances is the number of instances.

