# OpenReview forum: "Multiple Correlation Encoder-based Naive Bayes"
_ICLR.cc/2026/Conference — ICLR 2026 Conference Withdrawn Submission_

### Official Review · Reviewer_doFK · 2025-10-31

**Soundness:** 3
**Presentation:** 2
**Contribution:** 2
**Rating:** 2
**Confidence:** 4

**Summary:**

The paper proposes a two-stage learning framework that aims to address attribute–attribute, attribute–class, instance–instance, and instance–class correlations in naive Bayes.
The approach introduces three components:
(1) an ELBO-style objective for feature reconstruction,
(2) two regularization terms that minimize correlations between attribute-attribute, attribute–class, instance-instance, and instance–class correlations in naive Bayes,
(3) an additional linear weighting layer for final prediction.
Experiments are conducted on small-scale toy datasets to demonstrate the effect of each component.

**Strengths:**

1. The method is conceptually straightforward and easy to follow.

**Weaknesses:**

1. The overall motivation and contribution of the paper are unclear.
The proposed components resemble well-studied techniques in prior work.
Specifically, the ELBO objective is similar to VAE-regularized classification [1,2];
minimizing correlations among features has been explored in [3,4];
maximizing correlations between representations and classes is equivalent to optimizing a classifier via cross-entropy, as discussed in [2,5];
using contrastive learning to regularize classification has been explored in [6];
and the final weighting step is effectively a standard linear classifier trained with cross-entropy.
Thus, similar results could likely be achieved via an end-to-end MLP classifier trained with cross-entropy combined with a series of known regularizations [1,2,3,4,6], which would likely be simpler and computationally more efficient than the proposed two-stage framework.
As such, it is difficult to identify a novel technical contribution or conceptual insight.

2. The empirical evaluation is limited.
The experiments are performed only on small toy datasets with few features and limited samples, lacking validation on high-dimensional or real-world data such as images or text.
This makes it hard to assess whether the approach scales or provides meaningful improvements in realistic settings.
Moreover, the reported results exhibit large variances across datasets, suggesting unstable performance across runs.

3. There are several presentation issues.
The introduction claims to address attribute–class and instance–class correlations, but later sections appear to enhance certain correlations, which creates a conceptual inconsistency.
The paper also does not adequately explain why optimizing the ELBO is necessary or beneficial in this setting, and no ablation studies are provided to justify individual design choices.

- [1] Stainvas, Inna, Nathan Intrator, and Amiram Moshaiov. "Improving classification via reconstruction." Available at citeseer. nj. nec. com/article/steinvas00improving. html (2000).
- [2] Alemi, Alexander A., et al. "Deep Variational Information Bottleneck." International Conference on Learning Representations. 2017.
- [3] Cogswell, Michael, et al. "Reducing overfitting in deep networks by decorrelating representations." arXiv preprint arXiv:1511.06068 (2015).
- [4] Zbontar, Jure, et al. "Barlow twins: Self-supervised learning via redundancy reduction." International conference on machine learning. PMLR, 2021.
- [5] Poole, Ben, et al. "On variational bounds of mutual information." International conference on machine learning. PMLR, 2019.
- [6] Khosla, Prannay, et al. "Supervised contrastive learning." Advances in neural information processing systems 33 (2020): 18661-18673.

**Questions:**

Please refer to the weaknesses.

---

### Official Review · Reviewer_MDk3 · 2025-11-01

**Soundness:** 3
**Presentation:** 3
**Contribution:** 2
**Rating:** 4
**Confidence:** 2

**Summary:**

This paper introduces a unified Naive Bayesian framework to simultaneously model correlations among attributes, instances, and classes. The core of the proposed method is a multiple correlation encoder, which is optimized by two correlation objective functions (attribute and instance). This encoder provides a more comprehensive representation and consequently improves NB. The proposed method is validated through extensive experiments on public real-world and synthetic datasets, and achieves considerable performance.

**Strengths:**

1. The proposed method designs a multiple correlation encoder to jointly model correlations among attributes, instances, and classes.

2. The experimental results show that the proposed method obtains competitive performance.

3. This paper is written well and provides an in-depth analysis of effectiveness and efficiency.

**Weaknesses:**

1. The technical contributions of the proposed method are not clarified, especially compared with ICGNB. If I understand correctly, the major difference between the proposed method and ICGNB is how the encoder is learned: the proposed method utilizes the attribute correlation loss and the instance correlation loss, while ICGNB adopts the ELBO loss only. For the parts of attribute generation, augmentation, and weighting, it is hard to tell the difference between the two methods.

2. The selected baselines are out-of-date (mostly published from 2019 to 2023, except for ICGNB), and so are the related studies. Yet compared with ICGNB, the performance gains of the proposed method are marginal.

**Questions:**

1. The results of ICGNB reported in Table 2 are different from those in the original paper, while other methods remain the same. I wonder the reason behind such a gap.

2. The selected datasets are quite small; can the proposed method be evaluated on other large-scale attribute-related tasks, like zero-shot image classification?

---

### Official Review · Reviewer_S37T · 2025-11-01

**Soundness:** 2
**Presentation:** 3
**Contribution:** 2
**Rating:** 2
**Confidence:** 4

**Summary:**

The paper introduces an autoencoder-augmented variant of the Naive Bayes (NB) classifier, motivated by the idea that not only attribute-class relationships but also attribute-attribute, instance-class, and instance-instance dependencies should be considered in NB learning. The method operates in two main phases: first, an autoencoder is trained to learn latent features from the original inputs, and second, a weighted NB classifier is trained on the concatenation of both original and latent attributes. Class labels are injected into the latent representation to encourage class-dependent feature learning. The proposed loss function extends the standard ELBO objective by adding correlation-based regularization terms.

**Strengths:**

- The paper tackles an interesting question, whether integrating multiple forms of dependency into Naive Bayes models can enhance classification capability.
- The approach is clearly presented, and the implementation appears straightforward enough to reproduce. The structure of the paper, including figures and tables, contributes to readability.
- The motivation to move beyond the traditional "conditional independence" assumption of NB is conceptually sound and aligns with recent interest in hybrid generative-representation learning models.

**Weaknesses:**

- The central hypothesis that explicitly capturing four types of correlations will improve performance is not convincingly substantiated. The paper provides neither theoretical reasoning nor experimental evidence that isolating all four dependencies yields tangible benefits.

- Experimental validation is limited and not well targeted. Since the datasets used do not have known correlation structures, it is impossible to assess whether the proposed mechanism truly models the intended dependencies. A more principled design with synthetic or controlled datasets would have been necessary.

- The method's originality appears modest. Autoencoder-based extensions of Naive Bayes and weighted NB formulations have been explored in prior work; the present approach seems to combine these ideas rather than introduce a fundamentally new direction.

- Empirical results show only marginal or inconsistent gains over existing baselines. Furthermore, several baselines are somewhat outdated, which weakens the experimental claims.

- The analysis section is descriptive rather than analytical, the paper reports accuracy numbers but provides little interpretation of what the model actually learns or how the introduced components affect behavior.

- Overall, while the paper is competently executed, the technical novelty and empirical impact appear insufficient for acceptance at a top-tier venue like ICLR.

**Questions:**

- What is the rationale for concatenating both original and latent attributes? Does this indicate that the latent representation alone cannot fully encode the necessary information?

- Why must the dimensionality of the latent space be identical to that of the original features? Could a smaller or larger latent dimension provide better inductive bias?

- The loss function comprises multiple terms, yet the paper does not analyze their individual effects. Are all components effectively optimized? Reporting each loss term separately would clarify this.

- Were weighting coefficients between loss components considered or tuned? If not, why?

- Do the 24 datasets used in the experiments truly contain all four types of correlations? If not, how can the results be interpreted as evidence for the proposed approach?

---

### Official Review · Reviewer_yRcn · 2025-11-03

**Soundness:** 3
**Presentation:** 3
**Contribution:** 3
**Rating:** 4
**Confidence:** 3

**Summary:**

This paper presents a comprehensive approach to improving the Naive Bayes classifier by learning the correlations between features and instances ( (attribute-class, attribute-attribute, instance-class, instance-instance)). The core idea of using a learned encoder to generate new attributes that explicitly optimize multiple correlations is innovative and would have a better ability. The experimental design is thorough, using a substantial number of datasets and rigorous statistical tests.

**Strengths:**

1. Integrating data from both brain tissue and CSF provides a more holistic view of the AD proteome than most previous studies.
2. The systematic evaluation of four batch correction methods, using multiple metrics, is valuable.
3. The strategy of building models both with and without the core AD CSF biomarkers is excellent. This can demonstrate that the proteomic panels provide substantial predictive power independently.

**Weaknesses:**

1. The analysis is constrained by the proteins measured and the cohorts available.
2. The result section lacks deeper mechanistic insight. A pathway enrichment analysis or protein-protein interaction network analysis of the novel candidate biomarkers would have strengthened the biological narrative and proposed potential functional roles in AD pathogenesis.
3. This study lacks a true external validation on a completely independent cohort.
4. The study treats AD as a binary condition (Control vs. AD). It does not address disease subtypes or progression stages.

**Questions:**

1. The related works are too simple, and the authors do not analyze previous works in detail. The definition of the four types of correlation is unclear.
2. No external validation datasets.
3. Several symbols in  Fig. 1 are unclear. The details of the framework of MCENB are missing.
4. Maximizing L_A cannot ensure that each correlation (rho_bc and rho_bb) will be optimized.
5. Why combine original attributes with newly generated attributes? This could incur many redundancies.

---

### Note · Authors · 2025-11-22

I have read and agree with the venue's withdrawal policy on behalf of myself and my co-authors.